# Alarming Upward Trend in Multidrug-Resistant Bacteria in a Large Cohort of Immunocompromised Children: A Four-Year Comparative Study

**DOI:** 10.3390/cancers15030938

**Published:** 2023-02-02

**Authors:** Ana-Raluca Mihalcea, Nathalie Garnier, Cécile Faure-Conter, Nicolas Rama, Cécile Renard, Sarah Benezech, Yves Bertrand, Christine Fuhrmann, Carine Domenech

**Affiliations:** 1Institute of Pediatric Hematology and Oncology (IHOPe), 69008 Lyon, France; 2Hospices Civils de Lyon, 69002 Lyon, France; 3Centre Léon Bérard, 69008 Lyon, France; 4Immuno-Biology Lymphoma Laboratory, International Center of Research in Infectiology, INSERM U111-CNRS UMR5308, 69008 Lyon, France; 5Faculté de Médecine Lyon Est, Université Claude Bernard Lyon 1, 69008 Lyon, France; 6Institut des Agents Infectieux et de Pathologies Infectieuses, Hospices Civils de Lyon, 69008 Lyon, France; 7Faculté de Médecine et de Maïeutique Lyon Sud, Université Claude Bernard Lyon 1, 69921 Lyon, France

**Keywords:** immunosuppression, bacterial bloodstream infections, multidrug resistance, paediatric haematology and oncology, neutropenia

## Abstract

**Simple Summary:**

Infection is the second leading cause of death in patients with cancer. The emergence of multidrug resistant bacteria is an ongoing problem, leading to difficulties in the treatment of antibiotic resistant bacterial infections. Our main objective was to document the evolution of multidrug resistant bacteria in a tertiary centre in Lyon, France, which was initially stable between 2014 and 2017 in a study conducted by Raad C. et al. (2021). Following 2017, multidrug resistant bacteria seem to increase gradually in bloodstream infections and in digestive colonisations in a similar cohort in the same tertiary centre, considering that the antibiotic management did not change over the eight years. We want to raise awareness among health practitioners and to incite other similar centres to study their tendencies of multidrug resistant bacteria in order to avoid dangerous multidrug resistant infections in immunocompromised children with a rather good prognostic otherwise.

**Abstract:**

Documenting bacteremia at the onset of fever in immunosuppressed children is challenging; therefore, it leads to the early administration of broad-spectrum antibiotics. We aimed to analyse the evolution of antibiotic resistance profiles of bacterial bloodstream infections (BSI) and gut colonisations in a large cohort of immunocompromised children carrying a central venous catheter, in comparison with a prior, similar study conducted in our centre from 2014 to 2017. A retrospective, observational cohort study was conducted from January 2018 to December 2021, in a tertiary centre for paediatric immuno-haematology and oncology. Empirical antibiotic therapy was adapted to the immunosuppression risk group and prior bacterial colonisation. There was a mean of 6.9 BSI/1000 patient bed days. Multidrug-resistant bacteria (MDRB) associated BSI accounted for 35/273 (12.8%). The incidence of MDRB gum/gut colonisation and MDRB associated BSI increased annually and correlated with the level of immunosuppression (*p* = 0.024). One third (34.7%) of the BSI episodes were not associated with neutropenia. As compared to the previous study, an alarming emergence of MDRB responsible for gut colonisations and BSI in immunosuppressed children was reported over the last four years. The degree of immunosuppression directly correlates with the risk of having an MDRB gut colonisation or MDRB BSI.

## 1. Introduction

In recent decades, the outcome of immunocompromised children has greatly improved due to infection management and supportive care improvement [1,2]. However, as we gain experience with the use of therapeutic options in oncology patients, we maintain an increased use of antimicrobial agents. Documenting bacteraemia at the onset of fever in immunocompromised patients is, nevertheless, challenging and, therefore, leads to the early administration of empiric broad-spectrum antibiotics [3].

This heterogenous population is highly immunosuppressed owing to various conditions: congenital diseases such as severe combined immunodeficiency syndrome (SCID), malignancy-associated bone marrow dysfunction, myelotoxic chemotherapy or conditioning regimens required for hematopoietic stem cell transplantation (HSCT). For all these immunocompromised children, bacterial bloodstream infections (BSI) remain a leading cause of morbidity and mortality [4,5]. These BSIs account for nearly half of all nosocomial identified infections [6].

An infectious complication affects the quality of life as it requires hospitalisation, painful diagnostic procedures and, moreover, might be life-threatening in this immunocompromised population. In addition, the treatment involves costly antimicrobial compounds that may be associated with adverse reactions and drug-to-drug interactions. Moreover, an infectious episode may delay specific care (i.e., chemotherapy, radiotherapy or HSCT), adversely influencing the outcome of the underlying disease. Studies in the early 1970s established that cancer patients with febrile neutropenia had an increased risk of mortality if antibiotic therapy was held until infection was proven [7]. Over time, with the early initiation of empiric antibiotic coverage, the mortality rate for Gram-negative bacilli (GNB) bacteremia in neutropenic patients has fallen from 90% to 10% [7].

Antimicrobial resistance occurs over time and the rapid development of multidrug-resistant bacteria (MDRB) has become a major healthcare issue worldwide [8,9]. Thus, the objective of the present study was to assess whether or not our local practice could be associated with an emergence of MDRB.

In this study, we report four years of experience in our tertiary centre for paediatric immuno-haematology and oncology in Lyon, France. Our centre provides healthcare for patients with congenital immunodeficiency, as well as immunocompromised children due to prior chemotherapy for cancer (solid tumour, or hemopathy) and/or HSCT. Our main objective was to analyse the evolution of antibiotic resistance profiles of BSI and gut colonisations in immunocompromised child carriers of a central venous catheter in comparison with a prior similar study conducted in our centre from 2014 to 2017 [10].

## 2. Materials and Methods

### 2.1. Study Design and Participants

We conducted a retrospective, observational, monocentric cohort study from January 2018 to December 2021 in our institute where children and young adults (up to 23 years) are treated for haematological diseases, solid tumours and HSCT.

All patients treated in our centre and hospitalised for a BSI were eligible for this study. Episodes of BSI with fever but non-neutropenic were included alongside cases of febrile neutropenia. The criterion of exclusion was the absence of a central venous access device (CVAD) in order to maintain the homogeneity of antimicrobial protocols.

In our centre, stem cell transplantations were performed according to the guidelines of the French Society of Stem Cell Transplantation, which remained similar between the two study periods, 2014–2017 and 2018–2021. Most of the patients received a myeloablative conditioning regimen with total body irradiation and VP16 for acute lymphoblastic leukemia (ALL), or Busulfan/Cyclophosphamide for acute myeloid leukemia (AML) and other haematological diseases. The patients with ALL were treated according to the French national protocol CAALL—F01; those with AML were treated according to the international protocol, MyeChild. Patients suffering from malignant lymphomas and solid tumours were treated according to the current national or international ongoing protocols.

### 2.2. Risk Stratification of Immunocompromised Children

The immunosuppression level was stratified into four infectious risk groups [11]. The lower risk groups included patients with solid tumours and Hodgkin lymphomas for Group 1, and ALL, localised Burkitt lymphoma, lymphoblastic lymphoma and high-risk neuroblastoma for Group 2. Group 3 included patients with high-risk ALL, and high-risk Burkitt leukemias/lymphomas. The highest risk group, Group 4, included patients treated for AML, ALL relapses, allogeneic and autologous HSCT, aplastic anaemias and severe combined immunodeficiency (SCID).

### 2.3. Supportive Care

Our centre is divided into 2 different units (protected unit and conventional unit) with targeted isolation procedures. The highest isolation measures were taken for the most immunocompromised patients, Group 4. Our protected unit was previously described [12]. Raad et al. (2021) described decontamination protocols, and central line management in our centre in a previously published article [10]. Dietary diversity was broadened since 2021 in patients from Groups 2 and 3. Our patients are rarely admitted to other hospitals, and all episodes of fever are regularly admitted or transferred during the episode in our centre.

### 2.4. Data Collection and Definitions

Clinical data and microbiological records were retrospectively collected from children treated in our centre and hospitalised for a documented BSI during a febrile episode with or without neutropenia. We defined neutropenia as a neutrophil count less than 500 per mm^3^. We considered sepsis as a documented infection and haemodynamic instability responsive to fluid resuscitation and severe sepsis when there was a persisting hypotension despite fluid resuscitation and an indication of vasopressor therapy or transfer in the Intensive Care Unit (ICU). We carried out a retrospective cohort study based on the review of medical records of referred child patients. The patients’ legal guardians were not opposed to the report of their data. Two blood cultures (aero-anaerobic) were routinely drawn at each occurrence of fever. Blood cultures were only sampled centrally from the CVAD and not coupled with a peripheric blood culture, as the latter is often of poor quality and painful for children. For *Coagulase-negative Staphylococci (CoNS)*, a confirmed BSI was considered after 2 distinct positive blood cultures within 48 h. Blood cultures were processed by the routine diagnostic laboratory. The categorisation of bacteria sensitivity to antibiotics among susceptible, intermediate and resistant was carried out according to CASFM/EUCAST clinical breakpoints (http://www.eucast.org/clinical_breakpoints/ accessed on 1 August 2022). CVAD were inserted in the subclavian or jugular vein under general anaesthesia, mostly single lumen, after an antiseptic protocol using topical povidone–iodine. The CVAD were continuously surveyed during follow-up through a transparent bandage changed once every 10 days for a non-infected CVAD, brought to once every 5 days coupled with a topical antiseptic protocol (povidone–iodine) if superficial local infection was present. However, if significant purulence was present at the catheter exit site or presence of *Staphylococcus aureus*, *Pseudomonas* spp., *Stenotrophomonas* spp. and fungemia was observed, catheter removal was mandatory.

Gum and gut colonisation were examined using oral and rectal swabs (Eswab) weekly for patients belonging to Groups 3 or 4 and for patients previously treated in a foreign country. Patients in Group 2 had monthly screens, or weekly if a pathogen was identified. Bacterial cultures screened extended-spectrum *ß-lactamase Enterobacteriaceae*, nonfermenting GNB and oral *Streptococci* (groups A, C, G).

### 2.5. Empirical Intravenous Antibiotic Therapy Protocols

All patients with a CVAD and febrile neutropenia were empirically treated with a glycopeptide (vancomycin), aminoglycoside (amikacin) and a broad-spectrum penicillin adjusted to the risk group. Group 1 received ceftriaxone, Group 2 piperacillin–tazobactam, and Groups 3 and 4 piperacillin only in order to spare a large spectrum antibiotic therapy in these very immunocompromised paediatric patients who benefitted from gum and gut decontamination. Digestive decontamination in our centre was only provided for patients from Groups 3 and 4. They received mouth care 4 times a day and vancomycin mouthwashes (twice a day) to eliminate Streptococcus [13]. They also benefitted from a non-absorbable digestive decontamination by gentamycin (replaced by colimycin in case of presence of Pseudomonas) and amphotericin B. Patients from Groups 1 and 2 received only mouth care 4 times a day. Gut colonisation is defined as the presence of a microorganism in the digestive tract of a host, with growth and multiplication of the organism, but without interaction between the host and organism (no clinical expression, no immune response) [14]. We defined a probable translocation (PT) as the same bacteria with the same drug-resistance profile identified in the bloodstream as well as in the gum/gut colonisation screening for the same patient. However, no genetic data were conducted to confirm the bacteria’s translocation. We also adapted our empirical therapy to prior identified gum or gut bacterial colonisation and history of MDRB. MDRB are defined as bacteria that are resistant to one or more classes of antimicrobial agents [15]. An empirical carbapenem therapy was used for BSI identified in a patient with febrile neutropenia with mucositis, gastrointestinal pain and a history of gut colonisation with extended-spectrum ß-lactamase bacteria. If an infection was found, antibiotic therapy was secondarily tailored to the identified pathogen and the clinical and biological evolution. Vancomycin was stopped after 48 h if GPC were not identified. The aminoglycoside was stopped after 3 injections in the absence of bacterial documentation. The minimal required length of treatment was of 10 days for *CoNS* and 14 to 21 days for *Enterobacteriaceae, Pseudomonas aeruginosa* or *Staphylococcus aureus*, counting from the last sterile blood sample. Standard treatment protocols (chemotherapies and empiric antibiotics in case of febrile neutropenia) did not differ for any group over the 4 years of analysis [10].

### 2.6. Statistical Analysis

Descriptive statistics were reported in terms of absolute frequencies and percentages. Infection incidence density was calculated by dividing the number of events by the number of patient bed days and was expressed as the number of episodes/1000 patient bed days. Distribution of data was described in terms of the median and interquartile range (IQR). A Pearson Chi2 test or Fisher exact test was used for qualitative data comparison. For quantitative data, a Mann–Whitney test for two-group comparison or Kruskal–Wallis test for multiple group comparison was used. To investigate the interaction between factors associated with neutropenia, we performed a multivariate analysis using a regression model. Statistical significance was defined as *p* < 0.05.

## 3. Results

### 3.1. Patients’ Characteristics

A total of 273 BSI were identified in 196 patients, corresponding to a mean of 6.9 BSI/1000 patient bed days (Figure 1). BSI were distributed as follows: Group 1, 38 episodes/34 patients; Group 2, 95 episodes/66 patients; Group 3, 65 episodes/33 patients; Group 4, 97 episodes/63 patients. Patients’ characteristics are described in Table 1.

### 3.2. Bacteria Spectrum and Risk Group Distribution

Over four years, we identified a total of 295 bacteria which represented 254 (93%) monomicrobial BSIs and 19 (7%) polymicrobial BSIs. BSIs were mostly observed in Groups 2 and 4, with a stable percentage of infections per year in these groups over time (Table 1).

All our patients had CVADs; 201 tunnelled catheters (73.6% of BSI), 51 Porth-a-cath (PAC) (18.7% of BSI) and 21 PICCs (7.7% of BSI).

GPC accounted for 52.2% of all BSIs: 40% were documented with *CoNS*, 5.4% with *S. aureus* and 6.8% with *Streptococcus* (among which 1.7% were *Enterococcus*). None of the *S. aureus* were methicillin-resistant (*MRSA*). GNB accounted for 35.9% of all BSIs, including 24.1% *Enterobacteriaceae* (*Escherichia coli* (46.5%); *Enterobacter cloacae* (18.3%); *Klebsiella pneumoniae* (18.3%)) and 9.5% *Pseudomonas* spp. BSIs. The main bacteria causing BSI among all groups were *CoNS* and *Enterobacteriaceae*, with a proportion of 35.4% to 44.7% and 20% to 29.9%, respectively.

### 3.3. Polymicrobial Infections

We identified 41 episodes of polymicrobial infections in 19 patients (9.6%). No patient had more than one polymicrobial BSI. There was no statistical significance between the risk group and the occurrence of polymicrobial infections. Six out of nineteen patients with polymicrobial BSI did not have neutropenia during the BSI episode. Among all polymicrobial BSI, eighteen patients had tunnelled CVAD (8.8% of patients with tunnelled CVAD), three patients had PAC (6.1% of patients with PAC) and four patients had a PICCs (20% of patients with PICCs).

### 3.4. Incidence

The incidence of BSIs varied over time with an increase in the year 2020: from 5.3 episodes/1000 patient bed days in 2019 to 8.7 episodes/1000 patient bed days in 2020 (Table 1). There was no statistically significant difference between the distribution of types of bacteria among the years 2018–2021 (Figure 2). However, *Enterobacteriaceae*-related BSI increased gradually each year (1.3 BSI/1000 patient bed days in 2018 versus 2.3 BSI/1000 patient bed days in 2021), whereas *Staphylococcal* BSI and *Pseudomonas*-related BSI globally decreased since 2018 (Figure 3).

### 3.5. Neutropenic versus Non-Neutropenic Patients with BSIs

Documented bacterial BSI associated with febrile neutropenia represented 178/273 (65.2%) of patients. One third of our study population, i.e., 95/273 (34.8%) patients, did not have neutropenia prior to or during the episode of BSI. 

The distribution of bacteria was statistically different between the neutropenic and non-neutropenic groups (*p* = 0.024). On the one hand, there was a higher number of *Pseudomonas*-related BSIs in the neutropenic group (*p* = 0.013, OR 3.53 [1.18–14.6]). On the other hand, the proportion of *Staphylococcal* BSI was significantly higher in the non-neutropenic group than in the neutropenic group (*p* = 0.004, reciprocal of OR 0.49 [0.29–0.82]).

The level of immunosuppression (group risk) was statistically different between the neutropenic and non-neutropenic patients (*p* < 0.0001) (Figure 4). More than a half (68.4%) of the non-neutropenic BSIs occurred to patients from Groups 1 and 4.

The distribution of the type of CVAD among patients from the neutropenic group was not significantly different from the non-neutropenic group. The majority of patients in both groups had a tunnelled CVAD (78% of the neutropenic patients and 65% of the non-neutropenic patients). The CVAD was removed for 28.1% of the neutropenic patients and for 38.9% of the non-neutropenic patients.

### 3.6. Evolution of BSI and Sepsis

The majority of patients (84.6%) had a good outcome under antibiotic therapy. Sepsis occurred in 15.4% of the patients, among whom 5.4% were transferred to the ICU. The proportion of sepsis was higher in the neutropenic group; 9.9% versus 5.5% in the non-neutropenic group, but with no statistical value. The overall mortality was at 1.5% (four patients); three patients from the neutropenic group (1.7%) and one patient from the non-neutropenic group.

Eighty (84.2%) non-neutropenic patients had a simple evolution under antibiotic therapy, whereas fifteen (15.7%) patients had sepsis (seven GNB, seven GPC and one other bacteria) and three of them were transferred in the ICU (two GNB BSI and one GPC BSI). Only one death was registered in the non-neutropenic group (1.1% mortality), occurring four months after an allogeneic HSCT for severe immune deficiency in a patient who had a polymicrobial BSI with *Enterococcus faecium* and two MDRB GNB after faecal contamination of the CVAD.

In the neutropenic group, twenty-seven patients (15.2%) had sepsis (sixteen GNB, six GPC, one other bacteria and four polymicrobial BSI: one with two GNB and the other three BSI with at least one GPC). There were eleven transfers into the ICU (eight GNB, one GPC, one other bacteria and one polymicrobial BSI with GPC and other bacteria). All deaths were related to a polymicrobial infection, but were associated with either a fungal infection or a severe complication of their underlying disease. Sepsis was not considered responsible for all three deaths, but could not be overlooked.

### 3.7. Reinfections

Fifty-seven patients (20.9%) suffered from at least a second BSI. Forty-one patients underwent two BSIs; sixteen patients underwent three BSIs or more. The majority of the patients who became reinfected at least once was distributed evenly among Groups 2 to 4 (26.3% to 36.8%). Among these 57 re-infected patients, 35 had their CVAD removed once and nine had their CVAD removed twice. Among the patients infected twice, 12/41 patients were re-infected by *Staphylococcus* (*CoNS* and *S. aureus*).

All our patients had CVADs: 201 tunnelled catheters, 51 Porth-a-cath (PAC) and 21 PICCs. Eighty-seven CVAD (10 PAC vs. 68 tunnelled catheters vs. 9 PICCs) (31.9%) were removed, among which 18 (6.6%) were considered infected clinically.

Among 126 *Staphylococcal* BSI, 60 (47.6%) were associated with a reinfection, among which 40 were reinfections with the same type of *Staphylococcus* spp. Among these episodes of BSI, forty-five (75%) patients had tunnelled CVAD, eight (13.3%) patients had PAC and seven (11.7%) patients had PICCs. There was no statistical association between the type of CVAD and reinfection with a *Staphylococcus*.

### 3.8. Gum/Gut Colonisation and Probable Translocation

A total of 119 patients presented gum and/or gut bacterial colonisation. Eighty-seven gum and/or gut MDRB colonisations were identified in 64 patients. There was a positive correlation between the level of immunosuppression (from Group 1 with lowest level to Group 4 with highest level of immunosuppression) and the MDRB colonisation (*p* = 0.024) (Figure 5). The total number of gum and/or gut MDR colonisation increased annually from 2018 to 2021 (Figure 5). Five out of these eighty-seven MDRB were identified as Extensively Drug-Resistant (XDR) in three patients. All these three patients were only treated in our centre and at home (two of them had parenteral nutrition). One child from Group 4 was colonised with *Citrobacter freundii* XDR and was readmitted two months after with a different documented colonisation with *Klebsiella oxytoca* XDR. The two other children were colonised with *Enterobacter cloacae* XDR.

A total of 26/295 identified BSI were probably consecutive to gut translocations. Seven were drug-sensitive GNB and nineteen were MDRB GNB. Moreover, the amount of probable MDRB translocations increases with the level of immunosuppression (15/26 PT occurred to patients in Group 4). There were no patients from Group 1 who had a PT. Three out of the twenty-six PTs were associated with a non-neutropenic febrile episode.

### 3.9. Multidrug Resistant Bacteria

Thirty-five BSIs (12.8%) included at least one MDRB, with a total of 37 identified MDRB (two polymicrobial BSIs with two or more MDRB). The MDRB were mainly extended-spectrum ß-lactamase (36/38) and there were two XDR bacteria identified in two different BSIs: *E. cloacae* and *Achromobacter xylosoxidans*. MDRB GNB accounted for 33/37 identified MDRB. The use of carbapenems as an empiric antibiotic treatment was at 17.9% of BSIs. After bacterial identification, carbapenems were used in 12.8% of the total episodes of BSI (57.1% of which were MDRB BSI).

A total of 18/35 MDRB BSIs occurred in patients in Group 4. Within Group 4, 7/18 (38.8%) were non-neutropenic patients, all of whom were more than one month past an allogeneic HSCT. There was a correlation between the number of MDRB BSIs and group risk (*p* = 0.01, OR 2.4 [1.1–5.2]) (Figure 6). However, there was no difference in the incidence of GNB among all groups. Fifteen MDRB infections were caused by reinfections. All GPC were susceptible to vancomycin. We report no change over time in the resistance patterns of GNB.

## 4. Discussion

We report in our centre an alarming emergence of MDRB since 2018. Raad C. et al. conducted a previous similar observational study in our centre to analyse the trends of BSI and the emergence of MDRB infection in patients with febrile neutropenia from 2014 to 2017 [10]. Between 2014 and 2017, the authors included 186 patients with febrile neutropenia, representing a total of 310 BSIs. The incidence of BSIs declined over the four years, with a decrease in staphylococcal infections that we do not confirm in our study between 2018 and 2021. Moreover, while the proportion of MDRB BSIs remained consistently low throughout their study from 2014 to 2017, the incidence of MDRB in gut colonisation and in BSI in immunosuppressed children increased significantly from 2018 to 2021 The proportion of MDRB BSIs remained consistently low throughout their four-year study. While there had been no changes in the antibiotic stewardship since 2014, the incidence of MDRB in gut colonisation in immunosuppressed children increased significantly from 2018 to 2021. Moreover, both MDRB gut colonisation and MDRB BSI increased significantly with the level of immunosuppression. The concerning increase in MDRB BSI linked with the increase in MDRB gut colonisation highlights the issue of digestive translocation in immunosuppressed children. Neutropenia and chemotherapy-induced mucosal injury are known to be risk factors of translocation [16,17]; therefore, broad-spectrum penicillin adjusted to the risk group and gut colonisation is rapidly started. Our antibiotic protocol in case of fever was to use Piperacillin only for patients belonging to Groups 3 and 4, in order to spare a large spectrum antibiotic therapy and to avoid MDRB emergence in these very immunocompromised paediatric patients who benefited from a gum and gut decontamination. However, the results of our study, in comparison to the previous one, revealed a more important emergence of MRDB in this sub-group, driving us to use carbapenems in first line in these patients more frequently. When an episode of fever without neutropenia occurs in a patient carrying a CVAD, our practice is to start antibiotics covering only possible GPC BSIs [18,19]. Strikingly, over the last four years, one third of the non-neutropenic patients had a GNB BSI, among whom seven needed rapid fluid resuscitation, two were non-responsive afterwards and were transferred in the ICU, and there was one death, which occurred in the first days in the ICU. Lehrnbecher et al. reported similar results from a prospective study conducted in the last four years [20], whereas Sulis et al. found no serious complication in non-neutropenic children with bacteraemia in a retrospective study between 2009 and 2014 [21]. We hypothesize that broadening the dietary diversity in patients since 2021 in our centre might have played a role in the increased number of BSIs in non-neutropenic febrile patients. Periodic digestive colonisation screening should be performed in order to be aware of possible MDRB colonisation in lower risk groups, as well. Fortunately, the incidence of XDR bacteria remains stable over time [10]. Moreover, there was no MRSA identified, nor vancomycin resistant bacteria, over the four years.

Despite infection preventive strategies and empirical antibiotic therapy, the hazards of all BSIs did not change substantially from 2018 to 2021 and, thus, remain a major concern among immunocompromised children [22,23,24]. CVAD strategies have a certain impact on the incidence. In our study, ultrasonography was only used to document a possible septic thrombophlebitis in the case of local signs of infection at the catheter insertion site. However, in their study, Picardi et al. suggested a routinely performed early ultrasonography searching for possible septic thrombophlebitis in order to adapt to the antibiotic treatment and to remove the CVAD if necessary [25]. Therefore, it could be interesting to plan an ultrasound in a systematic way in patients with *CoNS* infections in order not to neglect a risk of later reinfection linked to an unrecognized septic thrombosis and of course in all patients presenting evocative local signs (pain, inflammation, etc.) that the clinician will have to look for carefully.

The results of our study also revealed a lower rate of BSI with PICCs than with classical tunnelled CVAD, as reported by Picardi et al. in their study on acute myeloid patients’ CVAD [26]. However, we also reported a few years ago that PICCS were associated with an increased risk of thrombosis in our paediatric patients with acute leukaemia [27]. Thus, although PICCS could prove be an interesting alternative to conventional tunnelled catheters in terms of infection risk, particular attention should be paid to the increased risk of thrombosis in children with leukaemia. *Staphylococcal* infection represents the majority among admissions for BSIs. *Enterobacteriaceae*-related BSIs are a major concern as their incidence increases gradually each year. We intend to raise a flag among health professionals faced with a rising trend of MDRB infections in a similar cohort of patients and following similar antibiotic therapy guidelines. In an era where the prognosis in paediatric oncology is continuously improving [28,29,30,31], toxic death should not become a leading cause of mortality because of our practice. As infection is the second leading cause of death in patients with cancer [32,33], our practice should ensure a constant sensitivity of pathogens in order to overcome this issue.

Our study has the limitations of a retrospective analysis. On the one hand, immunosuppressed children have multiple complications related to their disease/treatment or even other infectious complications such as fungal or viral infections. Therefore, the overall evolution of a bacterial BSI for each patient can be influenced not only by their level of immunosuppression but also by other associated complications. On the other hand, our centre is unique in France; it treats children and young adults for haematological diseases (malignancies, severe combined immunodeficiency), solid tumours and has a bone marrow transplantation unit. Hence, our study population correctly represents the general population of immunosuppressed children.

As chemotherapeutic agents and strategies are constantly improving, future studies should be conducted to analyse the impact of these changes on the immune system and, therefore, on the infectious risk. The risk of emergence of MDRB is still a major concern and should not be overlooked when studying antibiotic strategies. There is a need for knowledge improvement in defining and documenting digestive translocations.

## 5. Conclusions

The emergence of multi-resistant bacteria responsible for bloodstream infections and gut colonisations in immunosuppressed children was documented in the last four years in our centre. Although local microbiological ecology and antibiotic stewardship are directly related to our centre’s results, we documented an alarming upward trend of multidrug resistant bacteria in gum/gut colonisation of our immunocompromised patients, as well as in their bloodstream infection, in comparison with the previous study conducted in our centre from 2014 to 2017 in a similar paediatric cohort. The degree of immunosuppression directly correlates with the risk of having a MDRB gut colonisation or MDRB BSI.

## 6. Future Directions

Bloodstream infections remain a current issue among immunocompromised children with or without neutropenia. A failure to reduce the incidence of bacterial infection in our patients suggests that future infection prevention strategies should more effectively target BSI prevention in these high-risk patients. Interestingly, immunosuppressed patients with fever but non-neutropenic stay at risk of BSI with GNB. A prospective study should be conducted to evaluate the actual risk of infection in a non-neutropenic immunosuppressed patient.

## Figures and Tables

**Figure 1 cancers-15-00938-f001:**
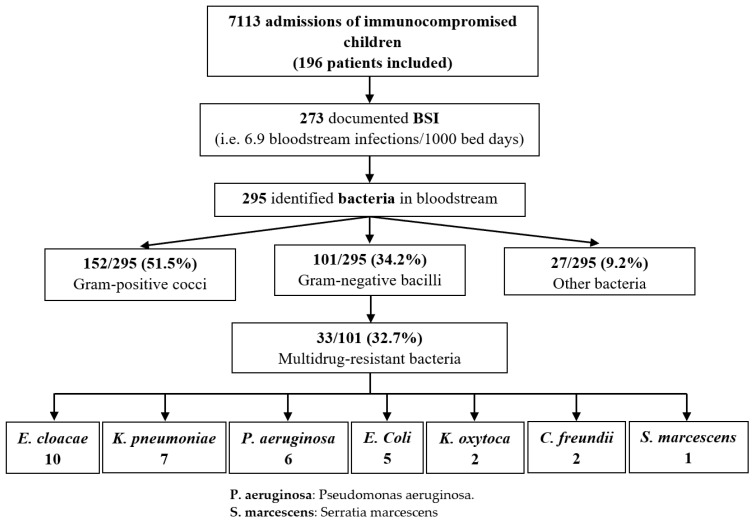
Bloodstream infections in immunocompromised children with or without febrile neutropenia: distribution of frequencies of MDRB.

**Figure 2 cancers-15-00938-f002:**
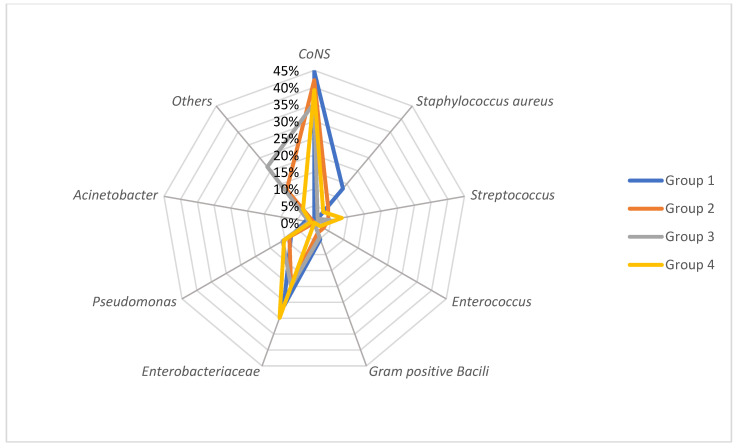
Comparison of the proportion of bacteria causing BSI in immunocompromised children by immunosuppression level group. Group stratification: Group 1 with a low level to Group 4 with the highest level of immunosuppression.

**Figure 3 cancers-15-00938-f003:**
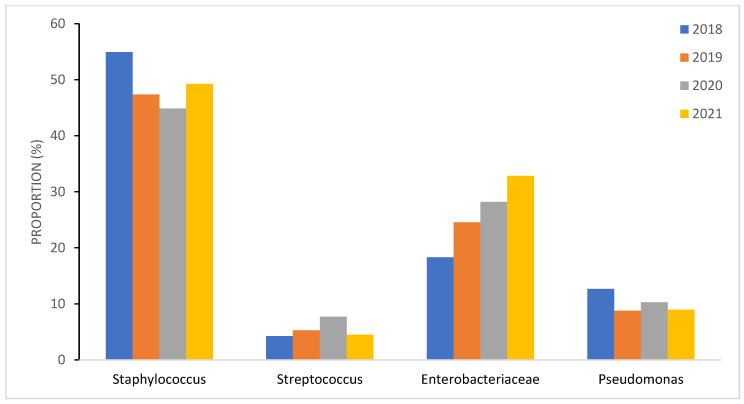
Four-year time distribution and evolution of the major pathogens causing BSI in immunocompromised children.

**Figure 4 cancers-15-00938-f004:**
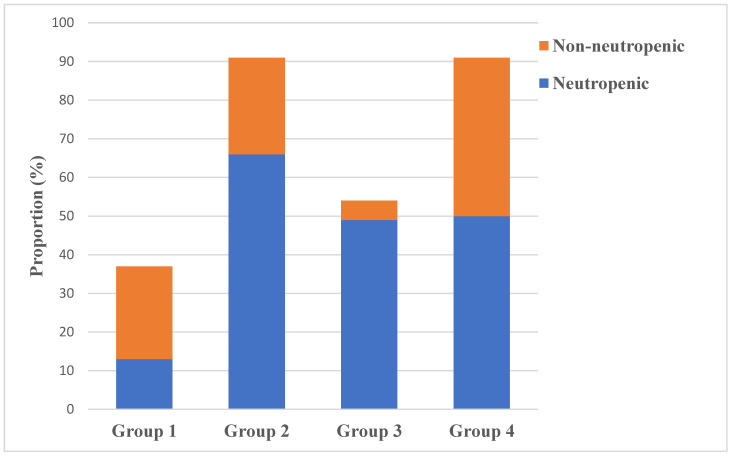
Distribution of neutropenic BSI and non-neutropenic BSI by group risk.

**Figure 5 cancers-15-00938-f005:**
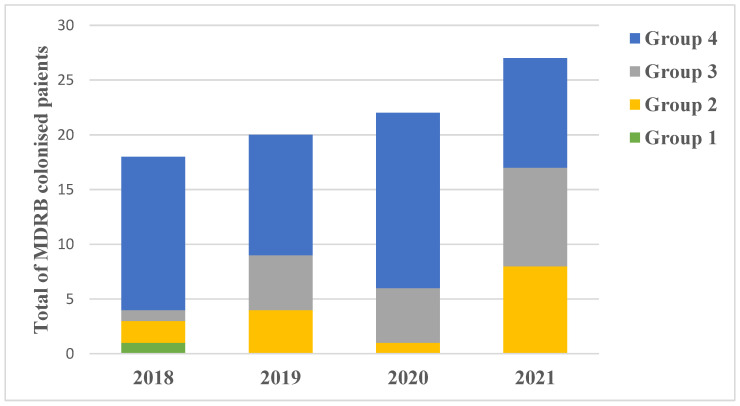
Annual distribution of the number of patients with gum/gut MDRB colonisation in each group of risk.

**Figure 6 cancers-15-00938-f006:**
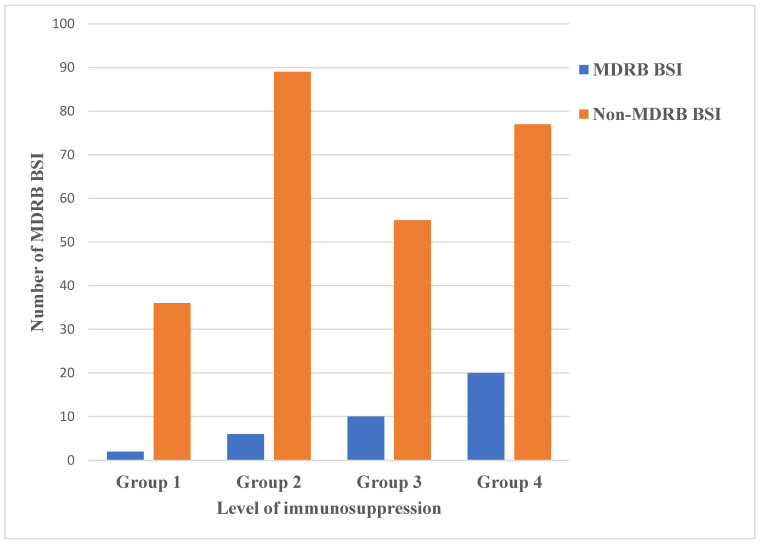
Distribution of MDRB and non-MDRB BSIs by level of immunosuppression.

**Table 1 cancers-15-00938-t001:** Bloodstream infections per year and per level of immunosuppression.

		2018	2019	2020	2021	4 Years
	Total admissions	1809	2129	1583	1592	7113
	Patient bed days	10,349	10,657	8953	9549	39,508
	BSI (%)	71 (3.9%)	57 (2.7%)	78 (4.9%)	67 (4.2%)	273 (3.8%)
	BSIs/1000 patient bed days	6.9	5.3	8.7	7	6.9
Group 1	Total BSIs	14 (19.7%)	8 (14%)	6 (7.7%)	9 (13.4%)	37 (13.5%)
Solid tumours	Age (years)(median-range)	6.5 (0–22)	14 (1–23)	5 (1–22)	7 (2–14)	7.5 (0–23)
Hodgkin malignant lymphoma	CVAD removal	3 (4.2%)	3 (5.3%)	2 (2.6%)	2 (3%)	10 (3.7%)
	Neutropenia	6 (8.5%)	2 (3.5%)	2 (2.6%)	3 (4.5%)	13 (4.8%)
	MDRB septicaemia	0	0	0	2 (3%)	2 (0.7%)
	Gum and gut MDRB colonisation	1 (1.4%)	0	0	0	1 (0.4%)
Group 2	Total BSIs	23 (32.4%)	16 (28.1%)	30 (38.5%)	22 (32.8%)	91 (33.3%)
ALL: low and medium risk groups	Age (years)(median-range)	5 (1–18)	7.5 (1–12)	13 (2–18)	7 (1–20)	9 (1–20)
Non-Hodgkin malignant lymphoma	CVAD removal	4 (5.6%)	5 (8.8%)	8 (10.3%)	9 (13.4%)	26 (9.5%)
Metastatic neuroblastoma	Neutropenia	17 (23.9%)	10 (17.5%)	22 (28.2%)	17 (25.4%)	66 (24.2%)
	MDRB septicaemia	0	0	2 (2.6%)	4 (6%)	6 (2.2%)
	Gum and gut MDRB colonisation	2 (2.8%)	4 (7%)	1 (1.3%)	8 (11.9%)	15 (5.5%)
Group 3	Total BSIs	9 (12.7%)	14 (24.6%)	15 (19.2%)	16 (23.9%)	54 (19.7%)
ALL: high risk group	Age (years)(median-range)	11 (3–18)	5 (1–19)	10 (3–18)	8 (1–19)	8 (1–19)
High risk Burkitt lymphoma/leukaemia	CVAD removal	4 (5.6%)	3 (5.3%)	3 (3.8%)	9 (13.4%)	19 (7%)
Non-Hodgkin malignant lymphoma high risk group/relapse	Neutropenia	9 (12.7%)	11 (19.3%)	15 (19.2%)	14 (20.9%)	49 (17.9%)
	MDRB septicaemia	2 (2.8%)	2 (3.5%)	1 (1.3%)	4 (6%)	9 (3.3%)
	Gum and gut MDRB colonisation	1 (1.4%)	5 (8.8%)	5 (6.4%)	9 (13.4%)	20 (7.3%)
Group 4	Total BSIs	25 (35.2%)	19 (33.3%)	27 (34.6%)	20 (29.9%)	91 (33.3%)
AML	Age (years)(median-range)	7 (0–19)	3 (0–15)	7 (0–19)	8 (0–19)	5 (0–19)
Acute leukaemia relapse	HSCT	16 (22.5%)	13 (22.8%)	19 (24.4%)	12 (17.9%)	60 (22%)
Medullary aplasia	CVAD removal	8 (11.3%)	8 (14%)	9 (11.5%)	7 (10.4%)	32 (11.7%)
Severe combined immunodeficiency syndrome (SCID)	Neutropenia	15 (21.1%)	10 (17.5%)	13 (16.7%)	12 (17.9%)	50 (18.3%)
HSCT (allogenic, autologous *)	MDRB septicaemia	4 (5.6%)	5 (8.8%)	5 (6.4%)	4 (6%)	18 (6.6%)
	Gum and gut MDRB colonisation	14 (19.7%)	11 (19.3%)	16 (20.5%)	10 (14.9%)	51 (18.7%)
ALL acute lymphoblastic leukaemia						
AML acute myeloid leukaemia						
HSCT haematopoietic stem cell transplantation						
MDR multi-drug resistant bacteria						

* During the first month after autologous HSCT.

## Data Availability

All the legal representatives of the children received a no-objection document at the initial diagnosis and we included only patients who agreed to participate in retrospective studies, in accordance with the MR004 law.

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
