# Peer review of "Alarming Upward Trend in Multidrug-Resistant Bacteria in a Large Cohort of Immunocompromised Children: A Four-Year Comparative Study"

_cancers, 2023, doi:10.3390/cancers15030938_

Round 1

Reviewer 1 Report

The study conveys an important clinical message based on data from the full spectrum of pediatric hematology and oncology and primary immunodeficiencies in childhood, at all phases of the disease and therapeutic levels of management, both for colonization and BSI. However, some conclusions might be context-dependent.

1. Management of fever in immunocompromised children in the authors' center is different from that used in the majority of centers according to guidelines (e.g. vancomycin-aminoglycoside without a beta-lactam in the absence of neutropenia; piperacillin without tazobactam in decontaminated patients). The possible link between those antimicrobial protocols and the observed increase in MDRB incidence should be discussed

2. Protocols of decontamination should be mentioned in the Methods section.

3. Type of CVAD should be reported in the Methods section or in Patients characteristics.

4. The title is too large. 

5. A formal comparison with the previous era (2014-2017) in the same center would be useful.

6. The use of carbapenems and new combinations of cephalosporins and beta-lactamase inhibitors for MDR would be useful.

7. Please correct 'medullary aplasia' by 'bone marrow aplasia'.

Author Response

(x) Moderate English changes required

To improve the quality of our English writing expression, we had our article revised by an English speaker (Mrs Raluca Anastasiu) and a native English speaker (Mr Steven Mead) that we acknowledged at the end of our manuscript.

  1. The study conveys an important clinical message based on data from the full spectrum of pediatric hematology and oncology and primary immunodeficiencies in childhood, at all phases of the disease and therapeutic levels of management, both for colonization and BSI. However, some conclusions might be context-dependent.

Response: We want to thank the reviewer for this remark. Indeed, our conclusions seemed to unwillingly generalise our results. Therefore, we adapted our conclusion in order to emphasize our centre‘s results regarding the upward trend of multidrug resistant bacteria in gum/gut colonisation of our immunocompromised patients as well as in their bloodstream infection in comparison to our previous study (2014-2017) conducted in our centre without any change of the local guidelines. We intend to incite other centres to study their local bacterial hospital ecology in order to have a better and a more accurate impact of a possible multidrug resistant bacteria emergence.

In-text proposal : Conclusion: “Although local microbiological ecology and antibiotic stewardship are directly related to our centre's results, we document an alarming upward trend of multidrug resistant bacteria in gum/gut colonisation of our immunocompromised patients, as well as in their bloodstream infection, in comparison with the previous study conducted in our centre from 2014 to 2017 in a similar pediatric cohort.”

  1. Management of fever in immunocompromised children in the authors' center is different from that used in the majority of centers according to guidelines (e.g. vancomycin-aminoglycoside without a beta-lactam in the absence of neutropenia; piperacillin without tazobactam in decontaminated patients). The possible link between those antimicrobial protocols and the observed increase in MDRB incidence should be discussed.

Response: We indeed need to emphasize more our centre’s related results in the text. Microbial monitoring of our hospital environment shows a majority of Coagulase Negative Staphylococcus infections, thus the frequent use of Vancomycin when a BSI is suspected. Therefore, we cover rapidly a possible Staphylococcal infection, but we de-escalate and stop the Vancomycin fast after bacterial documentation, thus reducing the risk of renal toxicity. This strategy also allows a faster discharge of patients with intravenous antibiotics thanks to our “At home hospitalisation” program. For the occurrence of severe Gram Negative Bacilli infections, we decided to keep using Amikacin for all patients. At last, our main objective to use Piperacillin only (vs Piperacillin-tazobactam) for patients belonging to groups 3 and 4 was to spare a large spectrum antibiotic therapy in order to avoid MDRB emergence in these very immunocompromised pediatric patients who benefited from a gum and gut decontamination.

In-text proposal : Discussion:  Our antibiotic protocol in case of fever  was to use Piperacillin only for patients belonging to groups 3 and 4 in order to spare a large spectrum antibiotic therapy and to avoid MDRB emergence in these very immunocompromised pediatric patients who benefited from a gum and gut decontamination. However, the results of our study, in comparison to the previous one revealed a more important emergence of MRDB in this sub-group, driving us to use carbapenems in 1st line in these patients more frequently.

and Conclusion (see comment 1)

  1. Protocols of decontamination should be mentioned in the Methods section.

Response: As asked by the reviewer, we added our protocols of decontamination in the Methods section.

In-text proposal : Materials and Methods section - Empirical intravenous antibiotic therapy protocols

“ Digestive decontamination in our centre was only provided for patients from groups 3 and 4. They received mouth care 4 times a day with the addition of vancomycin mouthwashes (twice a day) to eliminate Streptococcus (cf Brunet AS and al. Low incidence of sepsis due to viridans Streptococci in a ten-year retrospective study of pediatric acute myeloid leukemia. Pediatr Blood Cancer. 2006 Nov;47(6):765-72. doi: 10.1002/pbc.20706.) They also benefited from a non-absorbable digestive decontamination by gentamycin (replaced by colimycin in case of presence of Pseudomonas) and amphotericin B. Patients from groups 1 and 2 received only mouth care 4 times a day. ”

  1. Type of CVAD should be reported in the Methods section or in Patients characteristics.

Response: As proposed by the reviewer, we added the type of CVAD in our manuscript (results section: Patients’ characteristics and Polymicrobial infections).

In-text proposal : Results: Patients’ characteristics : “All our patients had CVADs: 201 tunnelled catheters (73.6% of BSI), 51 Porth-a-cath (PAC) (18.7% of BSI) and 21 PICCs (7.7% of BSI).”

AND Results: Polymicrobial infections : “Among all polymicrobial BSI, 18 patients had tunnelled CVAD (8.8% of patients with tunnelled CVAD), 3 patients had PAC (6.1% of patients with PAC) and 4 patients had a PICCs (20% of patients with PICCs).”

  1. The title is too large.

Response: We thank the reviewer for his /her comment. Therefore, we tried to propose a shorter title, more representative for our study.

Title proposal : Alarming upward trend in multidrug-resistant bacteria in a large cohort of immunocompromised children. A 4-year comparative study.

  1. A formal comparison with the previous era (2014-2017) in the same centre would be useful.

Response: We thank you for your proposal. Indeed, the previous manuscript was the starting point for this study, thus we agree that emphasizing the similarities and the differences between these 2 periods is very important.

In-text proposal : Discussion : “We report in our centre an alarming emergence of MDRB since 2018. Raad C. et al. conducted a previous similar observational study in our centre to analyse the trends of BSI and the emergence of MDRB infection in patients with febrile neutropenia from 2014 to 2017 [10]. Between 2014 and 2017, the authors included 186 patients with febrile neutropenia representing a total of 310 BSI. The incidence of BSIs declined over the 4 years with a decrease of staphylococcal infections that we do not confirm in our study between 2018-2021. Moreover, while the proportion of MDRB BSIs remained consistently low throughout their 4-year study, the incidence of MDRB in gut colonisation and in BSI in immunosuppressed children increased significantly from 2018 to 2021 as compared to 2014-2017 with however the same antibiotic stewardship from 2014 to 2021.”

  1. The use of carbapenems and new combinations of cephalosporins and beta-lactamase inhibitors for MDR would be useful.

Response: The use of carbapenems in probabilistic antibiotic treatment was at 17.9% of BSI. After bacterial identification, carbapenems were used in 12.8% of the total episodes of BSI (57.1% of which were MDRB BSI). The rest of 42.9% of the episodes where we used carbapenems was for germs naturally resistant, or as preferred treatment in terms of tissular diffusion. During this study period (2018 – 2021), we have not used new combinations of cephaloporins and betalactamase inhibitors for MDR in probabilistic or in curative therapy.

In-text proposal : Results: Multidrug resistant bacteria : “The use of carbapenems as empiric antibiotic treatment was at 17.9% of BSI. After bacterial identification, carbapenems were used in 12.8% of the total episodes of BSI (57.1% of which were MDRB BSI).”

  1. Please correct 'medullary aplasia' by 'bone marrow aplasia'.

Response: We thank the reviewer for his/her careful reading of the manuscript. We have made the requested modification in the manuscript.

Reviewer 2 Report

In this retrospective observational study, the Authors describe the changes in antimicrobial resistance profiles in blood culture over a 4-year period in a cohort of immunocompromised onco-hematological pediatric patients. Overall, the study is well conducted and clearly described. Data are of interest due to continous emergence of AMR different profiles and consider a particular category of patients and the potential impact on outcome. Some informations on immunosuppressive regimens could be of further interest in patient characterization. Some minor revision of Enghish languace could be considered to improve the manuscript.

Author Response

In this retrospective observational study, the Authors describe the changes in antimicrobial resistance profiles in blood culture over a 4-year period in a cohort of immunocompromised onco-hematological pediatric patients. Overall, the study is well conducted and clearly described. Data are of interest due to continuous emergence of AMR different profiles and consider a particular category of patients and the potential impact on outcome. Some information on immunosuppressive regimens could be of further interest in patient characterization. Some minor revision of English language could be considered to improve the manuscript.

Response : We thank the reviewer for his/her time spent to review our work and his/her useful comments to improve our manuscript. In our centre, stem cell transplantation is performed according to the guidelines of the French Society of Stem Cell Transplantation (SFGM-TC society). Most of patients received a myeloablative conditioning regimen with total body irradiation and VP16 for acute lymphoblastic leukemia (ALL), or Busulfan/Cyclophosphamide for acute myeloid leukemia (AML) and other haematological diseases.

The patients with ALL were treated according to the ongoing French national protocol CAALL – F01, those with AML were treated according to the international protocol: MyeChild. The chemotherapies for pediatric solid tumours were conducted in accordance with the international guidelines for each type of tumour.

Furthermore, to improve our English language, we had our article revised by an English speaker (Mrs Raluca Anastasiu) and a native English speaker (Mr Steven Mead) that we acknowledged at the end of our manuscript.

In-text proposal : Materials and methods : Study design and participants :In our centre, stem cell transplantations were performed according to the guidelines of the French Society of Stem Cell Transplantation that remained similar between the two study periods, 2014-2017 and 2018-2021. Most of the patients received a myeloablative conditioning regimen with total body irradiation and VP16 for acute lymphoblastic leukemia (ALL), or Busulfan/Cyclophosphamide for acute myeloid leukemia (AML) and other haematological diseases. The patients with ALL were treated according to the French national protocol CAALL – F01, those with AML were treated according to the international protocol: MyeChild. Patients suffering from malignant lymphomas and solid tumours were treated according to the current national or international ongoing protocols.”

Round 2

Reviewer 1 Report

After the amendments performed by the authors, the manuscript is suitable for publication, with all context-related issues being perceivable by the reader.